# Regulation of Hypoxic Signaling and Oxidative Stress via the MicroRNA–SIRT2 Axis and Its Relationship with Aging-Related Diseases

**DOI:** 10.3390/cells10123316

**Published:** 2021-11-26

**Authors:** Taku Kaitsuka, Masayuki Matsushita, Nobuko Matsushita

**Affiliations:** 1School of Pharmacy at Fukuoka, International University of Health and Welfare, Fukuoka 831-8501, Japan; kaitsuka@iuhw.ac.jp; 2Department of Molecular and Cellular Physiology, Graduate School of Medicine, University of the Ryukyus, Okinawa 903-0215, Japan; masayuki@med.u-ryukyu.ac.jp; 3Laboratory of Hygiene and Public Health, Department of Medical Technology, School of Life and Environmental Science, Azabu University, Sagamihara 252-5201, Japan

**Keywords:** sirtuins, SIRT2, hypoxia, HIF-1α, oxidative stress, microRNA, Parkinson’s disease, Alzheimer’s disease

## Abstract

The sirtuin family of nicotinamide adenine dinucleotide-dependent deacetylase and ADP-ribosyl transferases plays key roles in aging, metabolism, stress response, and aging-related diseases. SIRT2 is a unique sirtuin that is expressed in the cytosol and is abundant in neuronal cells. Various microRNAs were recently reported to regulate SIRT2 expression via its 3′-untranslated region (UTR), and single nucleotide polymorphisms in the miRNA-binding sites of SIRT2 3′-UTR were identified in patients with neurodegenerative diseases. The present review highlights recent studies into SIRT2-mediated regulation of the stress response, posttranscriptional regulation of SIRT2 by microRNAs, and the implications of the SIRT2–miRNA axis in aging-related diseases.

## 1. Introduction

The sirtuin family of nicotinamide adenine dinucleotide (NAD^+^)-dependent deacetylase and ADP-ribosyltransferases regulates health, aging, and life span. Some members of the sirtuin family participate in the stress response pathway, and the dysregulation of sirtuins is associated with aging-related diseases, such as type 2 diabetes mellitus, neurodegenerative diseases, various types of cancer, and cardiovascular disease [1]. Mammalian sirtuins consist of seven genes, SIRT1–7, with unique tissue and cellular distributions and participate in diverse cellular and tissue processes, such as nutrient metabolism, glucose homeostasis, circadian rhythms, and DNA repair [2,3]. Both SIRT1 regulation of the stress response pathway by sirtuins via hypoxia regulation and SIRT3 regulation of oxidative stress are well understood [4,5]. In these pathways, SIRT1 mainly deacetylates and activates hypoxia-inducible factor-2α (HIF-2α), which enhances the hypoxic response via induction of HIF-2α-dependent genes, whereas SIRT3 deacetylases and activates the tricarboxylic acid cycle enzyme, isocitrate dehydrogenase 2, which produces nicotinamide adenine dinucleotide phosphate (NADPH) in the mitochondria and has a protective effect against oxidative stress [4,5].

The cytosolic sirtuin SIRT2 also participates in the stress response caused by hypoxia and reactive oxygen species (ROS). Contrary to the actions of SIRT1 and SIRT3, SIRT2 is thought to repress the protective response and enhanced toxicity caused by hypoxia and oxidative stresses. Consistently, the inhibition of SIRT2 enhances the protective response against such stresses and protects cells, especially neuronal cells, in oxidative stress-related neurodegenerative disease models [6]. Therefore, the inhibition of SIRT2 is a candidate target for the treatment of neurodegenerative diseases, such as Alzheimer’s, Parkinson’s, and Huntington’s diseases.

MicroRNAs (miRNAs) are small noncoding RNAs that reduce the efficiency of the translation of target mRNA and participate in stress response pathways [7,8,9,10]. HIF and NF-E2 related factor 2 (Nrf2) upregulate or downregulate their expression in response to each stress condition. For example, in cancer cells, HIF-induced miRNAs translationally repress target mRNAs and influence tumor aggressiveness and metastasis [9,11]. Concerning the relation of miRNAs to sirtuins in those stress response pathways, many miRNAs are upregulated in response to hypoxic and oxidative stresses, and some of them participate in upstream and downstream signaling of sirtuin-regulated pathways. Sirtuins are directly regulated by miRNAs, and some have been identified and are described in the present review. Although the relationship between SIRT2 and miRNAs is gradually becoming clearer, it is still not well-understood.

The present review discusses recent advances in our understanding of the crosstalk between SIRT2 and miRNAs in stress response and summarizes these in terms of SIRT2-related diseases and aging.

## 2. Involvement of SIRT2 in Hypoxic Signaling and Oxidative Stress

The regulation of HIF proteins and hypoxic signaling by SIRT1 via NAD^+^-dependent deacetylation is well-understood, although there are conflicting reports about the mechanisms involved in SIRT1-mediated deacetylation of HIF-1α [12,13,14]. Deacetylation of HIF-1α at Lys674 by SIRT1 inactivates the protein in HEK293T cells [14], while another (unknown) deacetylation site leads to the stabilization and accumulation of HIF-1α in HeLa and Hep3B cells [12,13]. Therefore, it remains unclear whether SIRT1 activates or represses HIF-1α [15]. Dioum et al. reported that SIRT1 deacetylates HIF-2α at K385, K685, and K741 under hypoxic conditions [16]. Thus, SIRT1 transactivates HIF-2α, and its target genes, such as superoxide dismutase 2 and erythropoietin, were shown to be induced in cultured Hep3B cells and mouse kidneys.

The regulation of oxidative and hypoxic stresses by mitochondrial SIRT3 is also well-understood. SIRT3 suppresses the production of ROS, which are necessary for the hypoxic induction of HIF-1α; thus, SIRT3 prevents the induction of HIF-1α under hypoxic conditions [17]. Research into other sirtuins showed that SIRT4 prevented hypoxia-induced apoptosis in H9c2 cardiomyoblasts [18]. Furthermore, hypoxia induces miR-3677-3p, which represses SIRT5 expression and enhances the migration and invasion of hepatocellular carcinoma [19]. Yang et al. reported that overexpression of SIRT6 promotes HIF-1α expression by preventing its degradation via deubiquitination and reported that SIRT6 promotes the invasion, migration, proliferation, and tube formation ability of human umbilical vein endothelial cells (HUVECs) [20]. Interestingly, and in contrast to SIRT1 and SIRT3, SIRT7 negatively regulates both HIF-1α and HIF-2α protein levels via a mechanism independent of prolyl hydroxylation and proteasomal or lysosomal degradation [21]. Thus, the effects on HIFs differ among the sirtuin family members.

The relationship between SIRT2 and hypoxic signaling is less well-documented than that for SIRT1 and SIRT3 (Table 1). Lee et al. reported that the SIRT2 inhibitor AK-1 increased the ubiquitination of HIF-1α under hypoxic conditions, leading to HIF-1α degradation via a proteasomal pathway in A549 human lung cancer cells, HeLa cells, and HEK293 cells [22]. Conversely, Seo et al. reported that SIRT2 directly deacetylates HIF-1α at Lys709 and destabilizes the protein under hypoxic conditions in HeLa cells, and SIRT2 knockdown cells consistently showed high levels of HIF-1α [23]. The acetylation of HIF-1α at Lys709 increases its protein stability since this site is also a ubiquitination site, and deacetylated lysine inhibits ubiquitination [24]. We previously reported that SIRT2 knockout increased HIF-1α protein levels and induced its target genes, such as *VEGFA* and *LDHA*, in human and chicken B cells (Nalm-6 and DT40, respectively) [25]. Therefore, the effect of SIRT2 inhibition on HIF-1α remains unclear (Figure 1). In terms of the relationship between SIRT2 and other HIFs, Hu et al. reported that SIRT2 reduced the expression of *VEGFD* and lymphangiogenesis in hypoxia-induced head and neck cancer cells via deacetylation of HIF-2α and decreased transcriptional activity of HIF-2α-target genes [26]. Thus, SIRT2 appears to have a negative effect on HIFs and hypoxic signaling.

Krishnan et al. reported that HIF-1α transcriptionally suppresses *SIRT2* expression via the hypoxia response element in the *SIRT2* promoter in visceral white adipose tissue [27]. This was linked to suppression of β-oxidation via diminished deacetylation of peroxisome proliferator-activated receptor-γ coactivator 1α and repressed expression of β-oxidation and mitochondrial genes. In addition, HIF-2α was reported to activate the nicotinamide phosphoribosyl-transferase (NAMPT)–NAD^+^–SIRT axis in chondrocytes by upregulating NAMPT [28]. NAMPT is a rate-limiting enzyme in mammalian NAD^+^ biosynthesis that produces nicotinamide mononucleotide (NMN; a precursor of NAD^+^) from nicotinamide [29,30]. Thus, HIF-2α stimulates NAD^+^ synthesis and activates SIRT family members [28], which in turn promote HIF-2α protein stability and synergistically activate the NAMPT–NAD^+^–SIRT axis. SIRT2 and SIRT4 are positively associated with HIF-2α stabilization in the NAMPT–NAD^+^–SIRT axis in chondrocytes. Overall, SIRT2 expression may be transcriptionally regulated by HIFs, and a HIF–SIRT2–HIF axis has been shown to exist in some cell types.

## 3. Major miRNAs in Hypoxic Signaling and Oxidative Stress

### 3.1. miR-130a

miR-130a-3p is a hypoxia-induced miRNA, in which miR-130a-3p was named as solely miR-130a, while miR-130a-5p was miR-130a* [31]. In hypoxic conditions, miR-130a-3p expression is elevated and targets DEAD-box RNA helicase 6 (DDX6). Decreased levels of DDX6 enhance the translation of HIF-1α in an internal ribosome entry site element-dependent manner. Several studies have provided insight into the regulation of hypoxia signaling by miR-130a. Under hypoxic conditions, miR-130 represses tumor suppressor p21 (CDKN1A) expression and enhances hypoxia-induced smooth muscle proliferation [32]. In addition, miR-130a-3p potentially targets death-associated protein kinase 1 and suppresses ROS production and cell apoptosis induced by OGD in SH-SY5Y and mouse neuroblastoma cell line Neuro2a cells [33]. Furthermore, they showed that adeno-associated virus-expressed miR-130a-3p represses cerebral injury in an in vivo mouse model of perinatal hypoxic ischemic encephalopathy induced by middle cerebral artery occlusion/reperfusion (MCAO/R) [33]. However, Deng et al. reported opposite effects of miR-130a in MCAO models. They verified that miR-130a was increased in the brains of MCAO model rats and OGD-treated neurons [34]. In these animal models, suppressed miR-130a improved neurological function and alleviated nerve damage. In their models, miR-130a targeted X-linked inhibitor of apoptosis protein (XIAP), and the miR-130a–XIAP axis regulated neurological deficits in the MCAO model. These findings indicate that miR-130a-3p generally enhances the protective effects via upregulation of HIF-1α and the reduction of ROS against hypoxic and oxidative stress.

Regarding the involvement of miR-130a in sirtuin-related signaling and Parkinson’s disease, as mentioned above, the expression of miR-130a is highly regulated by hypoxia. Therefore, miR-130a may regulate metabolic systems, such as NAD^+^-dependent deacetylase sirtuins. Exosomal miR-130a-3p targets SIRT7 and leads to the downregulation of *SIRT7* expression and upregulation of Wnt signaling pathway-associated protein, thus promoting osteogenic differentiation of adipose-derived stem cells [35].

Interestingly, the laminin-511–Yes-associated protein 1 (YAP)–miR-130a pathway suppresses phosphatase and tensin homolog (PTEN) protein and protects midbrain dopaminergic neurons in Parkinson’s disease [36]. In this study, the authors suggested two protective pathways: (1) YAP1 activation by laminin-511 leads to increased expression of transcription factors for mid-brain dopaminergic neuron identity, such as LIM homeobox transcription factor 1α and paired-like homeodomain transcription factor 3 (PITX3), and (2) miR-130a induction by laminin-511–YAP leads to the suppression of the cell death-associated protein, PTEN. The laminin-511–YAP-miR-130a–PTEN pathway is suggested as a target for neuroprotective therapy for Parkinson’s disease.

### 3.2. miR-210

miR-210 is also well-studied as a hypoxia-induced miRNA, termed hypoxamiR [37]. Chan et al. reported that miR-210 targets iron-sulfur cluster assembly protein (ISCU1/2) and represses mitochondrial respiration and associated downstream processes in hypoxic conditions [38]. Recently, miR-210 was shown to simultaneously regulate the expression of multiple target genes in order to fine-tune the adaptive response of cells to hypoxia in the liver [39]. On the other hand, Liu et al. showed that miR-210-3p was upregulated by a mechanism dependent on the hypoxia-induced transcriptional activity of HIF-1α in the U87-MG glioma cell line [40]. Induction of miR-210-3p in glioma cells promotes the epithelial-to-mesenchymal transition via upregulation of transforming growth factor beta and induces chemoresistance, indicating a tumor-promoting role of miR-210-3p in glioma [40]. In addition, miR-210 expression is induced by ischemia and was also found to be upregulated in different ischemic diseases, such as hindlimb ischemia in mice and brain transient focal ischemia in rats [41,42,43,44]. Therefore, miR-210 regulates hypoxic and ischemic response in several tissues, targeting tissue-selective mRNA.

Concerning the involvement of miR-210 in sirtuin signaling and aging-related diseases, miR-210 is reported to induce microglial activation in neonatal hypoxic-ischemic encephalopathy (HIE) by partially targeting SIRT1 [45]. HIE is caused by oxygen deprivation to the infant brain, and neuroinflammation is a major contributor to brain injury in HIE, in which microglial M1 activation is involved. HIE upregulates miR-210 expression in microglia in the rat neonatal brain and induces activated microglia. The administration of anti-miR-210 suppresses microglia-mediated neuroinflammation and reduces brain injury [45]. Watts et al. reported that they identified 620 unique target genes of miR-210 in humans and showed significant enrichment of aging-related neurodegenerative pathways, including Huntington’s, Alzheimer’s, and Parkinson’s diseases [46]. They also validated that miR-210 directly regulates target genes, such as OXPHOS genes, *EIF4EBP1*, *VEGFB*, *MAP2K2*, and *APOE* [46].

### 3.3. miR-199

miR-199a-5p was identified to target the 3′-UTR of HIF-1α and repress its translation [47]. The miR-199a-5p-HIF-1α axis was reported to regulate the progression of several types of cancers. Zhong et al. showed that the overexpression of miR-199a-5p decreases cell proliferation and tumor angiogenesis in prostate adenocarcinoma (PCa) cell lines PC-3 and DU145 by targeting HIF-1α [48]. Recently, miR-199a-5p was found to induce cytotoxicity in oxygen-glucose deprivation and reperfusion (OGD/R)-injured H9c2 cells, a model of acute myocardial infarction, via the reduced expression of HIF-1α [49]. In addition, miR-199a-5p is reported to repress the proliferation, migration, and invasion of non-small cell lung cancer by suppressing HIF-1α and signal transducer and activator of transcription 3 [50]. Overall, miR-199a-5p inhibits cancer progression and promotes cytotoxicity by OGD.

To investigate the molecular mechanisms of sex differences in nicotine self-administration, Pittenger et al. performed an RNA-sequencing analysis on an array of miRNAs to be differentially regulated by nicotine. They found that the expression of miR-199a-5p and 214 are upregulated in female rats exposed to nicotine and SIRT1 is a common target of miR-199a-5p/214 involved in that sex difference [51]. Regarding the relation of miR-199a-5p with aging-related diseases, the inhibition of ID2-AS1, which is a sponge of miR-199a-5p, is reported to decrease cell death in MPP+-treated SH-SY5Y cells [52]. In their reports, the inhibition of ID2-AS1 led to the activation of miR-199a-5p, which targets IFNAR1, leading to the inhibition of JAK2/STAT1, thus alleviating neuronal injury in the Parkinson’s disease model.

### 3.4. miR-122

miR-122 is the most abundant mRNA in the liver, and its expression is affected by liver diseases, such as steatohepatitis [53]. In non-alcoholic fatty liver disease (NAFLD), miR-122 targets HIF-1α, vimentin, and mitogen-activated protein kinase kinase kinase 3, which regulate steatosis [53]. Long et al. reported that miR-122 is upregulated in NAFLD liver tissue, and free fatty acid-treated HepG2 and Huh-7 cells showed downregulation of SIRT1 and potentiated lipogenesis-related genes, such as *SREBP1*, *FASN*, *SCD1*, *ACC1*, and *APOA5* [54]. Furthermore, miR-122 suppressed SIRT1 expression by binding to its 3′-UTR. HIF-1α induced miR-122, which targets prolyl hydroxylase domain 1 during hepatic ischemia and reperfusion injury [55]. Therefore, HIF-1α appears to exist upstream and downstream of miR-122, suggesting that a negative feedback loop may regulate the miR-122–HIF-1α axis.

As mentioned above, miR-122 targets SIRT1, and this axis is reported to regulate chondrocyte extracellular matrix degradation in the cartilage of osteoarthritis [56]. The expression of miR-122 was increased in osteoarthritis cartilage compared to healthy controls, while *SIRT1* expression was decreased. The overexpression of miR-122 increased extracellular matrix catabolic factors, such as disintegrins and matrix metalloproteinases, and inhibited the expression of synthetic metabolic genes, such as collagen II and aggregating proteoglycan, partly via SIRT1 [56]. In the liver, miR-122 and SIRT6 negatively regulate each other’s expression, and they oppositely regulate a similar set of metabolic genes and fatty acid β-oxidation [57]. MiR-122 represses SIRT6 expression by binding to its 3′-UTR, while SIRT6 represses miR-122 expression by deacetylating H3K56 in its promoter region [57]. The possible involvement of miR-122 in Alzheimer’s disease was shown in an in vitro model using SK-N-SH cells treated with Aβ_25-35_ peptide and in an in vivo model of APPswe/PS1ΔE9 double transgenic mice [58]. Both lncRNA Rpph1 and miR-122 are up-regulated in that mouse model, and Rpph1 activates Wnt/beta-catenin signaling to ameliorate amyloid beta-induced neuronal apoptosis in SK-N-SH cells by directly targeting miR-122 [58].

## 4. Regulation of SIRT2 by miRNA and Its Relation to Cancer and Neurodegenerative Diseases

### 4.1. miR-212-5p–SIRT2 Axis

There are several predicted miRNA-binding sites in the 3′-untranslated region (3′-UTR) of the SIRT2 gene, and miRDB (3 October 2021, http://mirdb.org/mirdb/index.html) predicts that there are 53 miRNAs (e.g., miR-7155-5p, miR-1275, and miR-4283) that target SIRT2 [59,60]. Experimentally, SIRT2 is reported to be targeted by miR-212-5p (which is expressed from the 5′ end of the miR-212 precursor) in human colorectal cancer (CRC) via its 3′-UTR, leading to posttranscriptional downregulation of SIRT2 [61]. SIRT2 functions to suppress the proliferation and metastasis of the CRC cell line SW480, and the repression of SIRT2 by miR-212-5p promotes the proliferation and metastasis of SW480 cells [61]. SIRT2 regulation by miR-212-5p was also identified in pancreatic β cells through an exosome released from macrophages. Exosomal miR-212-5p released to pancreatic beta cells targets SIRT2 and regulates the protein kinase B (Akt)/glycogen synthase kinase-3β/β-catenin pathway, leading to the inhibition of insulin secretion [62]. SIRT2 was reported to bind with and activate Akt in the mouse fibroblast-like NIH3T3 cell line [63].

Sun et al. reported that miR-212-5p prevents 1-methyl-4-phenyl-1,2,3,6-tetrahydropyridine (MPTP)-induced death of dopaminergic neurons by inhibiting SIRT2 in a mouse model of Parkinson’s disease [64]. Furthermore, miR-212-5p was identified as an oncogenic miRNA (oncomir) [65] that is required for the invasion, migration, and growth of various cancers [66,67]. It is assumed that repression of SIRT2 is associated, at least in part, with miR-212-5p-mediated cancer cell progression, as supported by Du [61]. On the other hand, miR-212 is highly expressed in the brain, and its deregulation is associated with several brain-related diseases [68]. The expression of miR-212 is downregulated in α-synuclein A30P-transgenic mice, which is a mouse model of Parkinson’s disease [69]. The downregulation of miR-212 was also found in the cerebral cortical white matter of patients with Alzheimer’s disease [70], and it was recently reported that miR-212 is downregulated in neurally derived plasma exosomes of Alzheimer’s disease patients [71]. Thus, miR-212 has become an important therapeutic target and marker for the diagnosis of neurodegenerative diseases. Pharmacological enhancement of the function or expression of miR-212-5p could repress α-synuclein-induced neuronal death via downregulation of SIRT2. Xiao et al. showed that miR-212-5p prevents traumatic brain injury-induced ferroptosis, which is a newly discovered form of iron-dependent regulated cell death [72]. The authors reported that miR-212-5p prevented ferroptosis partially by targeting prostaglandin-endoperoxide synthase-2, although SIRT2 targeting may also be involved. Overall, the miR-212-5p–SIRT2 axis promotes cancer cell invasion, migration, and proliferation but was shown to prevent dopaminergic neuronal cell death in a Parkinson’s disease model [61,64].

### 4.2. miR-221-3p–SIRT2 Axis

In cardiomyocytes, miR-221-3p is expressed from the 3′ end of pre-miR-221 and targets the SIRT2 3′-UTR. Previously, Zhuang et al. showed that exosomes derived from mesenchymal stem cells pretreated with macrophage migration inhibitory factor (MIF) (exosome^MIF^) have a therapeutic effect against the chemotherapy drug doxorubicin (DOX)-induced cardiomyopathy [73]. Then they showed that the miR-221-3p–SIRT2 axis decreases the antisenescent effects of exosome^MIF^ in cardiomyocytes treated with DOX chemotherapy for cancer [73]. Furthermore, miR-221-3p is an oncomir and has been associated with pancreatic cancer [74]. Recently, miR-221-3p was shown to target HIF-1α in vascular endothelial cells and inhibit angiogenesis, and the inhibition of miR-221-3p improved the cardiac function of transverse aortic constriction-induced mice with heart failure [75]. Interestingly, Chen et al. reported elevated plasma levels of miR-221-3p in patients with early Parkinson’s disease and suggested it as a good biomarker for early Parkinson’s disease [76]. These reports support the relationship between miR-221-3p and SIRT2 in such diseases.

### 4.3. miR-140-5p–SIRT2 Axis

miR-140-5p is also reported to target SIRT2 and Nrf2. Zhao et al. reported that DOX induced cardiotoxicity by triggering myocardial oxidative damage, and miR-140-5p levels were increased by DOX in the rat cardiomyoblast H9C2 cell line [77]. They also reported that miR-140-5p directly targeted and downregulated Nrf2 and SIRT2, affecting the expression of heme oxygenase 1, NADPH quinone oxidoreductase 1, glutathione *S*-transferase, glutamate-cysteine ligase modifier subunit, kelch-like epichlorohydrin-associated protein 1, and forkhead box O3a and increasing DOX-induced myocardial oxidative damage [78]. The authors also showed that the natural compound dioscin alleviated DOX-induced cardiotoxicity via modulation of miR-140-5p [78]. Another study also showed that miR-140-5p targets SIRT2 and Nrf2 and regulates oxidative stress in HUVECs [79]. Song et al. showed that overexpression of miR-140-5p inhibited apoptosis of neurons under conditions of oxygen-glucose deprivation (OGD) in a model of ischemic stroke [80]. It is possible that the downregulation of SIRT2 by miR-140-5p is involved in this protective effect.

### 4.4. miR-7–SIRT2 Axis

It is reported that miR-7 targets Bcl2-associated X (Bax) and SIRT2 and inhibits the expression of SIRT2, leading to a decrease in RelA expression and relief of nuclear factor-kappa suppression, consequently protecting SH-SY5Y cells against MPP^+^ toxicity [81,82,83]. Li et al. showed that SIRT2 3′-UTR has the binding site of miR-7, and transfection with miR-7 decreased SIRT2 expression in SH-SY5Y cells [81]. In their model, MPP^+^ induces the enhancement of Bax and SIRT2 expression, leading to the expression of pro-apoptotic molecules, and miR-7 provides a protective effect by targeting these two mRNAs. Importantly, miR-7 targets α-synuclein mRNA and represses its expression and toxicity [84], highlighting miR-7 as a key factor in the pathogenesis of Parkinson’s disease. Consistently, the loss of miR-7 led to α-synuclein accumulation and dopaminergic neuronal loss in vivo [85], and miR-7 was considered a candidate therapeutic target for Parkinson’s disease [86]. These findings indicate that the therapeutic effect of miR-7 on Parkinson’s disease could involve the translational repression of SIRT2 and α-synuclein. 

### 4.5. miRNA–SIRT2 in Parkinson’s Disease (miR-486-3p, miR-376a-5p, and miR-8061)

In addition to the miR-212-5p–SIRT2 axis, the regulation of SIRT2 by other miRNAs in Parkinson’s disease has been reported. miR-486-3p binds to the 3′-UTR of SIRT2, and single nucleotide polymorphisms (SNPs), such as rs2241703, in this binding site were identified in Parkinson’s disease patients [87]. Another study also identified an SNP, rs2015, in the target site of miR-8061 in the 3′-UTR of SIRT2, which was shown to contribute to the risk of Parkinson’s disease [88]. The authors showed that treatment with miR-8061 mimic decreased SIRT2 protein levels in SH-SY5Y human neuroblastoma cells. Furthermore, both rs2015 and rs2241703 loci SNPs were shown to be associated with the risk of Alzheimer’s disease [89]. The same study showed that miR-376a-5p and miR-8061 bind to the rs2015 A allele to downregulate the expression of the SIRT2 protein. In terms of the function of these miRNAs, miR-486-3p is reportedly involved in the v-myb avian myeloblastosis viral oncogene homolog-driven control of erythroid versus megakaryocyte lineage fate decision [90]. miR-486-3p is also reported to play important roles in several types of cancers [91]. On the other hand, the functions of miR-8061 and miR-376a-5p are not well understood. Thus, miR-486-3p, miR3761-5p, and miR-8061 and their regulation of SIRT2 are attractive areas for future study in terms of their etiology and therapeutic use in Parkinson’s and Alzheimer’s diseases.

The miRNA-regulated repression of SIRT2 and their relationships with some diseases are summarized in Figure 2.

## 5. SIRT2, miRNAs, Hypoxia, Oxidative Stress, and Neurodegenerative Diseases

The repression of SIRT2 is well established to ameliorate neurological deficits in in vivo models of Alzheimer’s and Parkinson’s diseases. Wang et al. showed that a SIRT2 inhibitor rescued the cognitive impairment in amyloid precursor protein/presenilin 1 (APP/PS1) mice via the repression of Reticulon 4b deacetylation followed by the inhibition of beta-site amyloid precursor protein cleaving enzyme 1 [92]. In addition, the genetic reduction of SIRT2 rescued neuropathic phenotype in a *Drosophila* model of Charcot-Marie-Tooth disease type 2 caused by dominant mutations in *GARS*, which is a common inherited peripheral neuropathy [93]. Furthermore, the SIRT2 inhibitor AGK alleviated α-synuclein-mediated dopamine neuronal loss in vitro and in a *Drosophila* Parkinson’s disease model [94,95]. The pharmacological and genetic inhibition of SIRT2 also suppressed pathogenesis in *Drosophila* and mouse models of Huntington’s disease [96,97,98]. The involvement of SIRT2 in ischemic stroke was examined by Mori et al., who reported that SIRT2 protein levels changed in response to ischemia in the hippocampus of monkeys, indicating the involvement of SIRT2 in the ischemic pathway of primates [99]. In addition, the SIRT2 inhibitor AK-7 improves the outcome of brain ischemia independent of p38 activation in mice [100]. On the contrary, SIRT2 inhibition was reported to exacerbate traumatic brain injury [101,102]. AK-7 administration in mice treated with experimental traumatic brain injury increases the volume of brain edema lesion, neuroinflammation, and blood-brain barrier disruption via both increased K310 acetylation and nuclear translocation of NF-kappaB p65.

The inhibition of SIRT2 protects neuronal cells from oxidative stress, such as MPTP and hydrogen peroxide (H_2_O_2_). We previously reported that the SIRT2 inhibitor AGK2 reduces the H_2_O_2_-induced death of mouse cultured hippocampal neurons in a dose-dependent manner [25]. Consistently, Nie et al. found that AGK2 protected differentiated rat pheochromocytoma PC12 cells from toxic damage caused by H_2_O_2_ [103]. Another SIRT2 inhibitor, AK-7, ameliorated α-synuclein toxicity and showed neuroprotective effects in models of Parkinson’s disease [82,104]. In addition, the elevation of SIRT2 expression worsens motor impairment, while AK-7 treatment diminishes striatal dopamine depletion and improves behavioral abnormalities in rotenone-treated rats [105]. By contrast, Singh et al. showed that SIRT2 itself protected neural cells from oxidative stress; therefore, AGK2-mediated inhibition of SIRT2 promoted cell death [106]. However, although a few reports have shown contradictory effects, the inhibition of SIRT2 is thought to induce protective effects on neuronal cells against oxidative stress and α-synuclein toxicity.

SIRT2 is also known to influence autophagy and apoptosis mainly by regulating microtubule-related proteins and alleviating the toxicity of misfolded proteins, and this is well-reviewed by Chen et al. [107]. Specifically, Silva et al. showed that the inhibition of SIRT2 either by AK-1 or gene knockout restored microtubule stability and improved autophagy, favoring cell survival by eliminating toxic Aβ oligomers [108]. Regarding the involvement of the miRNA–SIRT2 axis in autophagy, miR-212-5p treatment prevents dopaminergic neuron loss by targeting SIRT2 as described above [64]. In this report, the authors showed that nuclear acetylated p53 is upregulated in consistent with the evidence that p53 is a major deacetylation substrate of SIRT2 [109], and decreased cytoplasmic p53 promotes autophagy in the Parkinson’s disease model [64].

## 6. Splicing Variants of SIRT2 and Another Posttranscriptional Regulation and Their Potential Relation to Neurological Diseases

The SIRT2 gene has three splicing variants: isoforms 1, 2, and 5. Isoform 1 is canonical SIRT2 that shuttles from the cytosol to the nucleus and shows deacetylase activity, whereas isoform 5 is constitutively localized at the nucleus and lacks deacetylase activity [110]. The differential expression of isoforms 1 and 2 is supposed to be regulated by their Kozak sequence surrounding each start codon [110]. Isoform 5 is expressed by exon-skipping of exons 2–4 [110].

Thangaraj et al. examined the posttranscriptional regulation of SIRT2 and showed that an RNA-binding protein, quaking (QKI), directly binds to the 3′-UTR of SIRT2 and stabilizes and extends the half-life of its mRNA [111]. The overexpression of QKI promoted the expression of SIRT2 mRNA and protein in CG4-OL cells derived from neonatal rat forebrain oligodendrocyte-type-2 astrocyte progenitors and regulated oligodendroglial differentiation [111]. QKI is a member of the signal transduction and activation of RNA family, which belongs to the hnRNP K-homology domain protein family [112]. QKI is expressed in the central nervous system during embryonic development [113] and is involved in human diseases, particularly neurological disorders, such as schizophrenia [114] and Alzheimer’s disease [115]. The involvement of SIRT2 in QKI-related neurological diseases is gaining increased attention. 

## 7. Therapeutic Strategy via miRNA–SIRT2 Inhibition

The development of microRNA-mediated therapies is underway for cancer and other diseases in the form of pri-miRNAs, anti-miRNAs, and miRNA mimics [116,117]. A phase I/II clinical trial for Huntington’s disease involving the delivery of pri-miR-451 via adeno-associated viral vectors has been reported [117]. Nanoparticle-based miRNA administration without viral vectors was recently developed for liver injury using nanoparticle-mediated miR122 overexpression via intravenous administration to a mouse ischemia/reperfusion injury model [55]. 

The administration of miRNAs, such as miR-212-5p and miR-486-5p, to target SIRT2 could be used as a therapeutic treatment for Parkinson’s disease and cancer. Sun et al. demonstrated that the stereotactic injection of miR-212-5p mimics into the midbrain of mice could prevent dopaminergic neuronal damage and loss [64]. The development of effective and safe administration methods of miRNAs into the striatum is desirable for the treatment of Parkinson’s disease.

Many SIRT2 inhibitors have been developed as artificial small molecules and natural compounds [118]. As discussed above, AGK2 and AK-7 showed therapeutic potential in an animal model of Parkinson’s disease. Another compound showing SIRT2 inhibition was found to have protective effects against neuronal cell death in a Parkinson’s disease model. ICL-SIRT078 showed a significant neuroprotective effect in a lactacystin-induced model of Parkinsonian neuronal death in the rat dopaminergic neural cell line N27 [119]. In addition, two compounds of 5-((3-aminobenzyl)oxy)nicotinamide derivatives exhibited significant protection against α-synuclein aggregation-induced cytotoxicity in the human neuroblastoma SH-SY5Y cell line [120]. A new class of SIRT2 inhibitors, *S*-trityl-L-cysteine, was identified to have potential therapeutic effects on cancer [121,122]. It would be interesting to evaluate the effect of these compounds in a model of Parkinson’s disease.

Recently, the delivery of antisense oligonucleotide (ASO) is developing for the therapy of Parkinson’s and Alzheimer’s diseases. A method using new indatraline (non-selective monoamine transporter inhibitor)-conjugated ASO (IND-ASO) was reported, as it disrupts the α-synuclein production selectively in monoamine neurons of a Parkinson’s disease-like mouse model and elderly monkeys [123]. Such ASO targeting of SIRT2 and microRNAs, which selectively acts on specific types of neurons, could be a desirable strategy for the therapy of neurodegenerative diseases.

## 8. Conclusions

The present review describes recent insight into the regulation of hypoxic and oxidative responses via sirtuins (especially SIRT2) and their regulation by miRNAs. Despite belonging to the same family, SIRT2 has opposite effects to SIRT1 in the stress response pathway, age-related diseases, and aging. Therefore, miRNAs targeting SIRT1 and SIRT2 could have opposite effects. The relationships between SIRT2 and Parkinson’s and Alzheimer’s diseases are becoming clearer, and SIRT2 inhibition is a crucial therapeutic strategy for these diseases. Thus, miRNAs targeting SIRT2 are desirable targets because these are natural endogenous factors; therefore, lipid-based nanoparticle-mediated delivery of miRNAs is a potential delivery mechanism for the treatment of neurodegenerative diseases. Furthermore, the presence of these miRNAs in peripheral blood mononuclear cells could be a good marker for preventing such diseases and promoting healthy aging.

## Figures and Tables

**Figure 1 cells-10-03316-f001:**
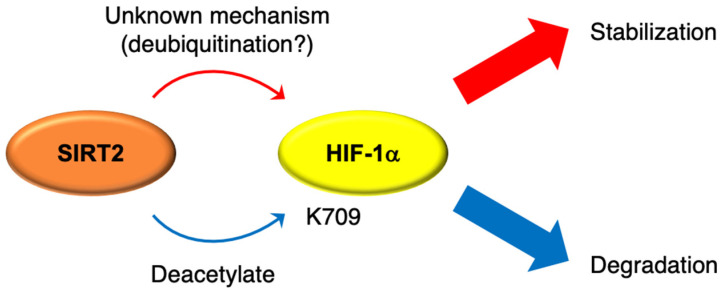
Schema of putative regulation mechanism of HIF-1α by SIRT2. The effects of SIRT2 on HIF-1α function remain unclear. Some studies support SIRT2 as an HIF-1α stabilizer, whereas others have shown that SIRT2 leads to degradation of HIF-1α via deacetylation at K709. HIF, hypoxia-inducible factor. K, lysine.

**Figure 2 cells-10-03316-f002:**
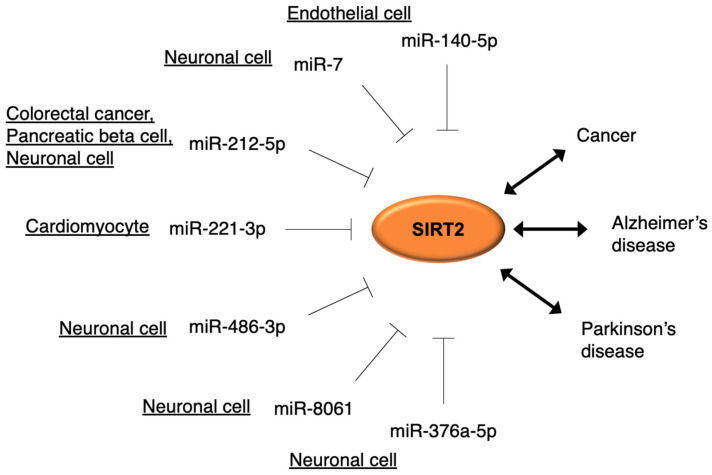
Summary of regulation of SIRT2 by miRNAs. miR-140-5p targets SIRT2 in HUVECs [79]. miR-7 targets SIRT2 and inhibits 1-methyl-4-phenylpyridinium (MPP^+^) toxicity in SH-SY5Y cells [81]. miR-212-5p targets SIRT2 and inhibits MPTP toxicity in mouse dopaminergic neurons [64]. miR-221-3p targets SIRT2 in cardiomyocytes [73]. miR-486-3p targets SIRT2 and SNPs of its binding site were found in Parkinson’s and Alzheimer’s diseases [87,89]. miR-8061 targets SIRT2 and SNPs of its binding site were found in Parkinson’s and Alzheimer’s diseases [88,89]. miR-376a-5p targets SIRT2 and SNPs of its binding site were found in Alzheimer’s disease [89].

**Table 1 cells-10-03316-t001:** Scientific reports identified using PubMed using indicated keywords.

Keywords	Hits	Plus Keyword	Hits
SIRT1	10,888	+microRNA	895
SIRT2	1676	+microRNA	30
SIRT3	1969	+microRNA	41
SIRT4	283	+microRNA	7
SIRT5	347	+microRNA	7
SIRT6	936	+microRNA	55
SIRT7	371	+microRNA	37
SIRT1 × hypoxia	424	+microRNA	57
SIRT2 × hypoxia	16	+microRNA	0
SIRT3 × hypoxia	103	+microRNA	4
SIRT4 × hypoxia	9	+microRNA	1
SIRT5 × hypoxia	7	+microRNA	1
SIRT6 × hypoxia	39	+microRNA	3
SIRT7 × hypoxia	9	+microRNA	3

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
