# Peer review of "Regulation of Hypoxic Signaling and Oxidative Stress via the MicroRNA–SIRT2 Axis and Its Relationship with Aging-Related Diseases"

_cells, 2021, doi:10.3390/cells10123316_

Round 1
Reviewer 1 Report
The Review “Regulation of hypoxic signaling and oxidative stress via the microRNA–SIRT2 axis and its relationship with aging-related diseases” by Kaitsuka and coworkers, refers the recent studies on SIRT2-mediated regulation of the stress response, posttranscriptional regulation of SIRT2 by microRNAs, and the implications of the SIRT2–miRNA axis in aging-related diseases.
The Review has been nicely performed, but minor revisions are required. In particular, I suggest scanning the text to remove some typos (repetitions, absence of italics where required etc). In addition, I suggest rearranging the order of some paragraphs. For instance, I wondered if paragraph 4 sounds better before the 3. Moreover, I guess that, discussing the same miRNA (mir-130a), paragraph 5 could be moved to paragraph 4.1. The same type of observation is for paragraph 6 which, in part, retakes some of the information discussed above.
Overall, after these minor revisions, I consider the manuscript ready for publication.
Author Response
We appreciate the reviewer’s suggestion. We have changed the orders of paragraphs as a composition of paragraph 5 is inserted to the section ‘4.1 mir-130a’ of paragraph 4 and paragraph 3 is replaced with paragraph 4. In addition, sentences about miR-7 in the paragraph 6 have been moved to section 4.4. miR-7-SIRT2 of this revised manuscript. We have read the manuscript thoroughly and revised some typos and changed typeface of gene name to Italic.
Reviewer 2 Report
- As mentioned in figure 2 several miRNAs are involved in the regulation of SIRT-2 in various disorders. Further, mir-130a has been discussed in sirtuin related signalling. However, miR-210, 199, 122 in relation to SIRT2 connected with ageing-related disorders should be discussed.
- Line 40- The cytosolic sirtuin, SIRT2, also participates in response to stresses caused by hypoxia and oxidants. Please write "reactive oxygen species" instead of oxidants.
- Line 52-54 sentence is unclear.
- “Regulation of SIRT2 by miRNA under stress and non-stress conditions” the title need to be changed. This part discusses the miRNA and diseased condition especially cancer and neurodegeneration.
- Line 138-spacing
- “Involvement of miR-130a in sirtuin-related signalling and Parkinson’s disease” Why this part is given as a separate heading. This section doesn’t discuss anything special to SIRT2. It’s a generalized section that can be combined with the previous section of miR-130a.
- Line 298-99- Modify the sentence
- In animal models of Traumatic brain injury, SIRT 2 selective inhibitor (AK- 7) disrupted BBB integrity and facilitated cerebral edema post-TBI. Further, SIRT 2 inhibition caused microglial activation, enhances NF-κB activation and cytokine production in TBI. (Yuan, F., Xu, Z.-M., Lu, L.-Y., Nie, H., Ding, J., Ying, W.-H., Tian, H.-L., 2016. SIRT2 inhibition exacerbates neuroinflammation and blood-brain barrier disruption in experimental traumatic brain injury by enhancing NF-κB p65 acetylation and activation. J. Neurochem. 136, 581–593. Ranadive, N., Arora, D., Nampoothiri, M. and Mudgal, J., 2021. Sirtuins, a potential target in Traumatic Brain Injury and relevant experimental models. Brain Research Bulletin) Similar contradictory results also may be included on SIRT-2 associated with brain injury.
- Line 321-330- The role of miR-7 in neurological disorders is understood. However, more focus should be given on the mechanism/how exactly it modulates SIRT2 to have a role in PD like conditions.
- “7. Splicing variants of SIRT2 and another posttranscriptional regulation” title is unclear.
- Line 335- The regulation of these isoform expressions is due to the exon-skipping or Kozak sequence. How does it regulate? References? Should be elaborated.
- What about the regulation of autophagy by SIRT2 and its impact on ageing and related disorders?
- “Therapeutic strategy via miRNA–SIRT2 inhibition” section is poorly developed. The section should be developed focusing on the current advances in therapy.
- “Link between miRNA and longevity via sirtuins” it’s a general section there is nothing specific to SIRT2. This section may be removed or reframed it should be connected with autophagy or other mechanisms specifically referring to SIRT2.
Author Response
- As mentioned in figure 2 several miRNAs are involved in the regulation of SIRT-2 in various disorders. Further, mir-130a has been discussed in sirtuin related signalling. However, miR-210, 199, 122 in relation to SIRT2 connected with ageing-related disorders should be discussed.
We appreciate the reviewer’s comments. We have searched literatures in which those miRNAs are shown to be related to sirtuin signaling and aging-related diseases and added some sentences into each section of 3.2 mir-210, 3.3 mir-199 and 3.4 mir-122 of this revised manuscript.
- Line 40- The cytosolic sirtuin, SIRT2, also participates in response to stresses caused by hypoxia and oxidants. Please write "reactive oxygen species" instead of oxidants.
We have changed ‘oxidants’ to ‘reactive oxygen species’ in line 41.
- Line 52-54 sentence is unclear.
We have re-written this sentence as ‘Concerning to the relation of miRNAs to sirtuins in those stress response pathways, many miRNAs are upregulated in response to hypoxic and oxidative stresses, and some of them participate upstream and downstream signaling of sirtuin-regulated pathways.’ in line 54-57 of this revised manuscript.
- “Regulation of SIRT2 by miRNA under stress and non-stress conditions” the title need to be changed. This part discusses the miRNA and diseased condition especially cancer and neurodegeneration.
We have revised this title to ‘Regulation of SIRT2 by miRNA and its relation to cancer and neurodegenerative diseases’.
- Line 138-spacing
We have added spacing to this portion.
- “Involvement of miR-130a in sirtuin-related signalling and Parkinson’s disease” Why this part is given as a separate heading. This section doesn’t discuss anything special to SIRT2. It’s a generalized section that can be combined with the previous section of miR-130a.
We agree with the reviewer’s comment. We have moved a composition of the section ‘5. Involvement of miR-130a…’ to the end of the section ‘3.1 mir-130a’ of this revised manuscript.
- Line 298-99- Modify the sentence
We have revised this sentence as line 574-576 of this revised manuscript.
- In animal models of Traumatic brain injury, SIRT 2 selective inhibitor (AK- 7) disrupted BBB integrity and facilitated cerebral edema post-TBI. Further, SIRT 2 inhibition caused microglial activation, enhances NF-κB activation and cytokine production in TBI. (Yuan, F., Xu, Z.-M., Lu, L.-Y., Nie, H., Ding, J., Ying, W.-H., Tian, H.-L., 2016. SIRT2 inhibition exacerbates neuroinflammation and blood-brain barrier disruption in experimental traumatic brain injury by enhancing NF-κB p65 acetylation and activation. J. Neurochem. 136, 581–593. Ranadive, N., Arora, D., Nampoothiri, M. and Mudgal, J., 2021. Sirtuins, a potential target in Traumatic Brain Injury and relevant experimental models. Brain Research Bulletin) Similar contradictory results also may be included on SIRT-2 associated with brain injury.
We have added sentences about the exacerbating effect of SIRT2 inhibitor on TBI as contradictory result to line 584-588 of this revised manuscript.
- Line 321-330- The role of miR-7 in neurological disorders is understood. However, more focus should be given on the mechanism/how exactly it modulates SIRT2 to have a role in PD like conditions.
We have added some sentences about the mechanism which the reviewer pointed out to line 386-389 of this revised manuscript.
- “7. Splicing variants of SIRT2 and another posttranscriptional regulation” title is unclear.
We have changed the title to ‘Splicing variants of SIRT2 and another posttranscriptional regulation and their potential relation to neurological diseases’.
- Line 335- The regulation of these isoform expressions is due to the exon-skipping or Kozak sequence. How does it regulate? References? Should be elaborated.
We have revised the corresponding sentences and added reference as reviewer suggested at line 617-619.
- What about the regulation of autophagy by SIRT2 and its impact on ageing and related disorders?
We have added some sentences about the regulation of autophagy by SIRT2 to line 602-611 in this revised manuscript.
- “Therapeutic strategy via miRNA–SIRT2 inhibition” section is poorly developed. The section should be developed focusing on the current advances in therapy.
We have added some sentences about current therapy and expected strategy to line 727-733 in this revised manuscript.
- “Link between miRNA and longevity via sirtuins” it’s a general section there is nothing specific to SIRT2. This section may be removed or reframed it should be connected with autophagy or other mechanisms specifically referring to SIRT2.
We agree with the reviewer’s suggestion. We have removed this section.